# Deciphering the Role of Post-Translational Modifications and Cellular Location of Hepatitis Delta Virus (HDV) Antigens in HDV-Mediated Liver Damage in Mice

**DOI:** 10.3390/v16030379

**Published:** 2024-02-28

**Authors:** Sheila Maestro, Nahia Gomez-Echarte, Gracian Camps, Carla Usai, Cristina Olagüe, Africa Vales, Rafael Aldabe, Gloria Gonzalez-Aseguinolaza

**Affiliations:** 1DNA & RNA Medicine Division, Centro de Investigación Médica Aplicada, University of Navarra, Avenida Pío XII, 31008 Pamplona, Spain; smaestro@alumni.unav.es (S.M.); ngomez.9@alumni.unav.es (N.G.-E.); gcamps@unav.es (G.C.); carla.usai@irta.cat (C.U.); colague@unav.es (C.O.); avales@unav.es (A.V.); 2IdiSNA—Instituto de Investigación Sanitaria de Navarra, 31008 Pamplona, Spain

**Keywords:** HDV, HDAg post-translational modification, isoprenylation, phosphorylation, ribozyme, intracellular localization: cell and mouse models, liver damage

## Abstract

Hepatitis D virus (HDV) infection represents the most severe form of chronic viral hepatitis. We have shown that the delivery of HDV replication-competent genomes to the hepatocytes using adeno-associated virus (AAV-HDV) as gene delivery vehicles offers a unique platform to investigate the molecular aspects of HDV and associated liver damage. For the purpose of this study, we generated HDV genomes modified by site-directed mutagenesis aimed to (i) prevent some post-translational modifications of HDV antigens (HDAgs) such as large-HDAg (L-HDAg) isoprenylation or short-HDAg (S-HDAg) phosphorylation; (ii) alter the localization of HDAgs within the subcellular compartments; and (iii) inhibit the right conformation of the delta ribozyme. First, the different HDV mutants were tested in vitro using plasmid-transfected Huh-7 cells and then in vivo in C57BL/6 mice using AAV vectors. We found that Ser177 phosphorylation and ribozymal activity are essential for HDV replication and HDAg expression. Mutations of the isoprenylation domain prevented the formation of infectious particles and increased cellular toxicity and liver damage. Furthermore, altering HDAg intracellular localization notably decreased viral replication, though liver damage remained unchanged versus normal HDAg distribution. In addition, a mutation in the nuclear export signal impaired the formation of infectious viral particles. These findings contribute valuable insights into the intricate mechanisms of HDV biology and have implications for therapeutic considerations.

## 1. Introduction

Hepatitis Delta Virus (HDV) is the smallest known virus that infects humans. It belongs to the Kolmoviridae family and the Deltavirus genus [1]. It is a satellite virus that relies on the surface proteins of the Hepatitis B Virus (HBsAg) for assembly, hepatocyte entry, and the release of infectious particles [2]. HDV virions consist of HBV surface antigens and host cell lipids surrounding a ribonucleoprotein (RNP) complex formed by a single-stranded circular RNA genome of about 1700 nucleotides, along with hepatitis delta antigens (HDAg) [2,3,4]. Upon hepatocyte infection, the HDV RNA genome translocates into the nucleus, where the host cell RNA polymerase initiates replication through a rolling-circle mechanism [5,6,7]. This replication strategy allows the production of linear concatemers of HDV genomes and antigenomes that are self-cleaved by the HDV ribozymes, which are intrinsic catalytic RNA domains present in each RNA strand [8,9]. The antigenomic RNA encompasses a single open reading frame that encodes the short hepatitis delta antigen (S-HDAg) [10]. Through editing by the host cell’s adenosine deaminase acting on RNA-1 (ADAR-1) at the UAG stop codon, an amber (W) site is formed. This modification results in an extended open reading frame (ORF) that gives rise to the large hepatitis delta antigen (L-HDAg) [10]. Despite sharing most of their amino acid sequence, both the short and the large antigens differ significantly in their functions. S-HDAg is essential for HDV replication, whilst L-HDAg blocks HDV replication and is essential for viral assembly [11,12]. Both HDAgs contain a nuclear localization signal (NLS) located between positions 66 and 75 (EGAPPARAR), being Glu-66 (E66) and Arg-75 (R75), indispensable for the nuclear import of HDV RNP [13,14,15]. The L-HDAg contains nuclear export signal (NES) and viral assembly signal (VAS) domains [16] and is responsible for the translocation of HDV RNPs from the nucleus to the cytoplasm and viral assembly. The NES is located within aa 198–210 of the L-HDAg sequence. More specifically, Pro-205 is the critical residue for the correct functionality of NES [16].

The association between HBsAg and HDV RNP takes place in the ER via a lipid farnesyl moiety present on the L-HDAg that binds to the cytoplasmic loop of the small HBV surface antigen (S-HBsAg) and is indispensable for promoting viral spreading and completing the HDV life cycle [17,18]. Furthermore, phosphorylation of some residues of S-HDAg plays an important role in viral replication [19]. More specifically, phosphorylation of Ser177 promotes HDV replication by increasing the interaction of S-HDAg with RNA Pol II [19,20,21,22]. During the HDV replication, S-HDAg co-localizes with the host RNA pol II in the nucleoplasm, within the SC35 speckles sites that are highly active sites of transcription and RNA processing [7,23]. In earlier steps, S-HDAg is predominantly located in the nucleus, both in the nucleolus and the nucleoplasm. Later in the infection, when L-HDAg starts to be synthesized, the HDAgs can be found in non-SC-35 speckles sites, in the cytoplasm, and in the Golgi apparatus, where the post-translational modifications take place [24,25,26]. Thus, the correct localization of HDAgs is an important aspect of the HDV viral cycle.

HDV infection is recognized as the most severe form of viral hepatitis. HDV-infected patients have a higher risk of developing cirrhosis and hepatocellular carcinoma (HCC) as well as hepatic decompensation and increased mortality in comparison with HBV mono-infected patients [27]. Despite the severity of this disease, the underlying mechanism is still unknown and hence there is a lack of effective treatments to control HDV-induced liver damage [28]. One of the main reasons for the scarce knowledge of the molecular mechanism/s involved in the pathology of this disease is the absence of adequate animal models that resemble the main pathological features observed in HDV patients and are amenable to experimentation [29]. Recently, we utilized adeno-associated vectors (AAVs) as delivery vehicles for HBV/HDV replication-competent viral genomes. The co-administration of recombinant AAV-HBV and AAV-HDV to WT mice resulted in the establishment of AAV-independent HDV replication and, more importantly, the animals developed significant significant liver pathology characterized by transaminase elevation, lobular inflammation, cytoplasmic swelling, and sanded nuclei, which was not observed in AAV-HBV mice [30,31,32] and which mimics most of the features of severe acute HDV infection in humans [33,34,35]. Thanks to the easy manipulation of the system, we were able to confirm in vivo what had been previously shown in HDV cell culture models, i.e., that S-HDAg is essential for HDV replication and cannot be replaced by L-HDAg or host cellular proteins, and that L-HDAg is essential to produce the HDV infectious particle and inhibits its replication [31]. Furthermore, we found that the L-HDAg/S-HDAg ratio plays an important role in the magnitude of HDV-induced liver damage: the lower it is, the stronger the liver damage [31].

The aim of the present work was to identify the role of post-translational modifications and cellular location of HDV antigens in the HDV viral cycle and HDV-mediated liver damage. We found that Ser177 phosphorylation and ribozymal activity are essential for HDAg expression and HDV replication. Mutations in the NES and NLS affect HDAg intracellular localization and HDV replication in vivo. More importantly, mutations of the isoprenylation domain prevent the formation of infectious particles and reduce the L-HDAg/S-HDAg, increasing cellular toxicity both in vitro and in vivo.

## 2. Materials and Methods

### 2.1. Site-Directed Mutagenesis (SDM)

Site-directed mutagenesis (SDM) was performed using the TaKaRa In-Fusion Cloning Kit (Takara Bio, Otsu, Japan), except for the HDV-NPrL-HDAg plasmid, which was carried out using QuikChange II Site-Directed Mutagenesis. In brief, using the TaKaRa In-Fusion Cloning Kit, four oligonucleotides (Table 1) were designed for each HDV mutant, except for the HDV-ΔHDAg mutant, which required six oligonucleotides (Table 1), since it was necessary to introduce two mutations. Then, 10 ng of the HDV plasmid was used for the amplification of the new mutants. In parallel, the plasmid was digested with the appropriate restriction enzymes. The empty vector obtained after the enzymatic digestion and the PCR products were purified and ligated for 15 min at 50 °C. Subsequently, Stellar competent cells were transformed by heat shock, and the obtained clones were sequenced to verify the presence of the desired mutations and the absence of unspecific mutations. When using QuikChange II Mutagenesis, two primers were designed to produce the HDV-NPrL-HDAg mutant as recommended in the QuikChange^™^ manual (Agilent, Santa Clara, CA, USA #200524). For the PCR amplification, 50 ng of the HDV plasmid, 125 ng of primer Forward (Fw), 125 ng of Primer Reverse (Rv), 1 μL of dNTP mix, 3 μL of QuikSolution, and 1 μL of PfuUltra High Fidelity (HF) DNA polymerase (Agilent, #600380) were mixed at a final volume of 50 μL and PCR-amplified. Then, 10 μL of the PCR product was run in an electrophoresis gel to check the quality of the reaction. To remove the methylated DNA plasmid (in this case, the non-mutated HDV plasmid), 1 μL of the DpnI restriction enzyme was added to the reaction product and was incubated at 37 °C for 1 h. Finally, the PCR product was electroporated using One Shot^®^ Top10 Electrocomp^™^ *E. coli* (Thermo Fisher Scientific Inc., Waltham, MA, USA #C404052). Plasmid DNA was purified using the NucleoSpin^®^ Plasmid kit (Macherey-Nagel, Düren, Germany #740588), and the correct plasmids were identified by restriction enzyme digestion and sequenced to verify the introduction of the correct mutation and to discard unspecific mutations.

### 2.2. Cell Lines

The human hepatoma cell line Huh-7 and the human embryonic kidney fibroblast cell line HEK293-T were acquired from Glow Biologics, Tarrytown, NY, USA (GBTC-099H) and ATCC, Manassas, VA, USA (CRL-3216), respectively. Huh-7 cells were used for the HDV plasmid transfection studies and 293-T cells for AAV vector production. Huh-7.5.1 stably expressing the human Na-taurocholate co-transporting polypeptide (hNTCP, Huh-7-hNTCP), which is essential for HBV and HDV cell entry [36,37], were kindly provided by Dr. Urtzi Garaigorta and were employed for infectivity studies. Both cell lines were cultured in Dulbecco’s modified Eagle’s medium (DMEM) supplemented with 10% fetal bovine serum (FBS) (DMEM 10% FBS), 1% of L-glutamine, 1% of glucose, 100 U/mL of penicillin-streptomycin and non-essential amino acids and incubated at 37 °C with 5% CO_2_ in a humidified atmosphere. In the case of Huh-7-hNTCP, DMEM was supplemented with 2.5 μg/mL of blasticidine to ensure the selection of hNTCP-expressing cells.

### 2.3. DNA Transfection

For transfection, Huh-7 cells were used at a confluence of 80–90%. On the day of transfection, the culture medium was replaced by Opti-MEM™ (Gibco, Thermo Fisher Scientific Inc., Waltham, MA, USA #31985070). In brief, Lipofectamine 3000 (Thermo Fisher Scientific, Inc., Waltham, MA, USA # L3000001) and Opti-MEM™ were mixed, vortexed, and incubated for 5 min at room temperature (RT) (reaction mix A). In parallel, plasmid DNA, the P3000 reagent, and Opti-MEM™ were also mixed, vortexed, and incubated for 5 min at RT (reaction mix B). Then, mixes A and B were mixed, without vortex, and incubated for 15 min at RT. After that time, the transfection mix was added drop by drop. Six hours after adding the transfection reagents, fresh DMEM 10% FBS was added to the culture plates without removing the medium. Finally, 1 day after transfection, the culture medium was removed, and the cells were washed before adding fresh DMEM 10% FBS.

### 2.4. Generation of AAV Vectors

HEK293T cells were used as packaging cells. The plasmids containing the recombinant AAV genomes were transfected together with the helper plasmid pDP8.ape (Plasmid Factory, Bielefeld, Germany #PF478), which provides the genes required for the replication and encapsidation of AAV serotype 8 (a liver tropic AAV serotype in mice). The transfection was carried out using poly-ethyleneimine (PEI) (Sigma-Aldrich, Merck KGaA, Darmstadt, Germany #408727). The cells were incubated for 72 h and during this time the viral particles were assembled inside the packaging cells. The cells were collected in lysis buffer (50 mM Tris, 150 mM NaCl, 2 mM MgCl_2_, 0.1% Triton X-100), and various freeze–thaw cycles were performed to release the viral particles. Cell debris was removed by centrifugation. The supernatant of the cells was collected and incubated in 8% polyethylene glycol (PEG-800) Sigma-Aldrich, #P5416) for 48–72 h to precipitate the viral particles. The supernatant was centrifuged, and the pellet was resuspended in a lysis buffer. Cell lysate and the precipitated supernatant were mixed and treated with 0.1 mg/plate of DNaseI (Roche, Basel, Switzerland #10104159001) and RNaseA (Roche, #10109169001) for 1 h at 37 °C. The viral particles were purified by ultracentrifugation in an iodioxanol gradient (OptiPrep™ Merck KGaA, Darmstadt, Germany #D1556). Iodixanol was removed and the vector was concentrated using Amicon 100 K columns (Sigma-Aldrich, Merck KGaA, Darmstadt, Germany, #UFC910008). Viral DNA was extracted using the High Pure Viral DNA Kit (Roche, #11858874001) following the manufacturer’s instructions. The vector titer in terms of viral genomes (vg) per mL was calculated by real-time qPCR (RT-qPCR) (BioRad, Hercules, CA, USA #1855196) using specific primers for the AAV’s inverted terminal repeat (ITR) sequences.

### 2.5. Animal Manipulation and Procedures

WT C57BL/6 mice were purchased from Harlan Laboratories (Barcelona, Spain). Six- to eight-week-old male mice were used in all experiments. The mice were kept under controlled temperature, light, and pathogen-free conditions in a BSL3 animal facility.

The animals received the vectors intravenously, diluted in saline solution (Braun, Barcelona, Spain #651860) in a final volume of 100 μL/mouse under inhalatory anesthesia. All mice received 5 × 10^10^ vg of the corresponding AAV-HDV WT or mutant vector together with 5 × 10^10^ vg of AAV-HBV.

The experimental design was approved by the Ethics Committee for Animal Testing of the University of Navarra (R-132-19GN).

### 2.6. Protein Extraction from Cells and Liver Samples

RIPA Buffer (0.75 M NaCl, 5% of Tris 1 M pH 8, 0.1% SDS, 1% Triton X-100, and 0.5% sodium deoxycholate, diluted in water) was supplemented before each use with 1 mM sodium orthovanadate, 1 mM PMSF, 1 mM sodium pyrophosphate, and a protease inhibitor cocktail. Cell pellets were resuspended and incubated for 30 min at 4 °C on a shaker with the RIPA lysis buffer and centrifuged at 13,000 rpm for 20 min at 4 °C. Supernatants were collected in new tubes. Liver pieces were homogenized by mechanical disruption, using RIPA buffer. After 30 min of incubation at 4 °C in a shaker, liver samples were ultracentrifuged at 75,000 rpm for 20 min at 4 °C in a Hitachi Micro Ultracentrifuge CS 150NX. Supernatants were collected in new tubes. Protein concentration was calculated using the Pierce BCA Protein Assay Kit (Thermo Fisher Scientific, #23225).

### 2.7. Immunofluorescence (IF)

Cells were washed once with PBS, fixed with cold 4% paraformaldehyde (PFA) (freshly prepared from methanol-free 16% PFA, Thermo Scientific, #28908) for 15 min at RT and washed again three times with PBS. For intranuclear staining, cells were permeabilized with 0.1% Triton X-100 in PBS for 15 min at RT and, after three washes with PBS, a blockade was performed with 5% BSA in PBS-0.1% tween for 30 min at 37 °C. Next, the cells were incubated for 30 min at 37 °C with anti-HDV human serum provided by the BioBank of the Universidad de Navarra (CUN-28336) at dilution 1:2500. After washing three times, the secondary antibody (anti-human Alexa-Fluor 488,) was incubated for 30 min at 37 °C diluted 1:3000 in PBS and protected from light. Finally, coverslips were mounted on microscope slides using a mounting medium with DAPI for fluorescent labeling (Vector Laboratories, #H-1200, Newark, CA, USA). Fluorescent samples were obtained with a confocal microscope (Zeiss LSM 880 NLO, Oberkochen, Germany) at 40×, 60×, and 100× of magnification.

### 2.8. Determination of HDV Infectious Viral Particles

Huh-7-hNTCP cells seeded in BD Falcon^TM^ 8-well Culture Slides (BD Biosciences, San Jose, CA, USA, #345118) or a 6- or 12-well plate with a coverslip were incubated for 24 h with HDV-containing cell supernatants. The supernatants were then removed, and the cells were washed with fresh PBS. Then, the cells were maintained with a fresh medium that was changed every 2 days until 7 days post-infection. At this time point, cells were incubated for 15 min at RT with 4% PFA, and HDAg expression was assessed by immunofluorescence, as described above.

### 2.9. Western Blot

20–30 μg of extracted proteins were mixed with an SDS-PAGE loading buffer (63 mM Tris HCl 30% glycerol, 0.35 M SDS, 0.6 M DTT, 0.18 mM Bromophenol Blue) and boiled for 5 min at 95 °C. Samples were loaded on 12% SDS-polyacrylamide gels of 1.5 mm and electrophoresis was performed for separating proteins. 5 ul of Precision Plus Protein™ Dual Color Standard (BioRad, #161-0394), ranging from 10 kDa to 250 kDa, was used to determine protein size. Subsequently, the resolved proteins were transferred to a nitrocellulose membrane (Bio-Rad #162-0112) by wet electroelution at 80 V for 1 h and 15 min. Following that, the membrane was stained with Ponceau Red dye to check the transference quality. After washing the dye with water, the membrane was blocked for 45 min with Tris-buffered saline (TBS) with 0.1%Tween 20 (TBS-T) with 5% of non-fat dry milk at RT on a shaker. Primary antibodies were added and incubated ON while shaking at 4 °C. Next, the membrane was washed 3 times with TBS-T for 10 min and then it was incubated with the appropriate horseradish peroxidase (HRP)-conjugated secondary antibody at RT for 1 h. After 3 washes of 10 min, the peroxidase signal was developed using SuperSignal^TM^ West Femto Maximum Sensitivity Substrate (Thermo Fisher Scientific, #34095). Odyssey CLx near-infrared fluorescence imaging system was used for image generation and images were analyzed with the Image Studio Lite software v5.5.

### 2.10. Histology and Immunohistochemistry (IHC)

Hematoxylin and eosin (H&E): Liver sections were fixed with 4% PFA, embedded in paraffin, sectioned (3 μm), and stained with hematoxylin and eosin. Sections were mounted and analyzed by light microscopy for histologic evaluation.

Immunohistochemistry (IHC): the first steps were the same as for the H&E staining. Then, a step of antigen retrieval was performed that consisted of incubation for 30 min at 95 °C in 0.01 M Tris-1 mM EDTA pH 9. Subsequently, primary antibodies were incubated overnight at 4 °C. After rinsing in TBS-T, the sections were incubated with the corresponding secondary antibodies for 30 min at RT. Peroxidase activity was revealed using DAB+, and sections were lightly counterstained with Harris hematoxylin. Finally, slides were dehydrated in a graded series of ethanol, cleared in xylene, and mounted with Eukitt (Labolan, #28500, Navarra, Spain). Image acquisition was performed with an Aperio CS2 slide scanner using ScanScope Software 12.3 (Leica Biosystems, Vista, CA, USA). The image analysis was performed using a plugin developed for Fiji, ImageJ (NIH, Bethesda, MD, USA). The antibodies employed were an anti-HDV human serum sample (CUN-28336) and anti-F4/80 (BioLegend, San Diego, CA, USA #123102).

### 2.11. RNA Extraction and RT-qPCR

The total RNA from liver samples was isolated using TRI Reagent^®^ (#T9424, Sigma-Aldrich,) according to the manufacturer’s instructions. The total RNA was pre-treated with DNAse I (#AM-1907, TURBO DNA-free ™ Kit, Applied Biosystems, Waltham, MA, USA) and reverse-transcribed into complementary DNA (cDNA) using M-MLV reverse-transcriptase (Invitrogen, Carlsbad, CA, USA #28025013). RT-qPCR was performed using iQ SYBR Green Supermix (#170-8884, BioRad) in a CFX96 Real-Time Detection System (#1855196, BioRad) and primers as specified in Supplementary Materials. HDV strand-specificity was analyzed as described elsewhere [30]. GAPDH was used as a control housekeeping gene.

### 2.12. Statistical Analysis ç

Statistical analyses were performed using GraphPad Prism 8.0 software. The data are presented as individual values ± standard deviation. The statistical analysis performed in each experiment was specified in the legend of the figures (significance * *p* < 0.05, ** *p* < 0.01, *** *p* < 0.001, **** *p* < 0.0001 and ns, non-significant).

## 3. Results

### 3.1. Analysis of HDAgs Expression and HDV Replication in Huh-7 Cells Transfected with HDV WT and Mutants

We constructed the mutants described in Figure 1A as follows: an HDV mutant expressing a non-prenylated L-HDAg (HDV-NPrL-HDAg), where Cys-211 was replaced by a serine as described by Glenn et al. [17]; two mutants altering HDAg phosphorylation, one non-phosphorylated with Ser177 substituted by alanine, and another with Ser177 replaced by the phosphomimetic amino acid aspartate [19,20,21]; an HDV mutant expressing HDAgs lacking the nuclear localization signal (NLS), with Glu-66 and Arg-75 substituted by Ala (HDAg-ΔNLS) [15]; an HDV mutant expressing L-HDAg without the nuclear export signal (NES), where Pro-205 was replaced with Ala-205 [16]; and finally, we introduced a mutation in helix 1 of the antigenomic HDV ribozyme (HDV-ΔRibozyme). More specifically, the nucleotide sequence CCGG located at position 691–694 was replaced by AGCC, which resulted in a conformational change in the ribozyme [9].

The Huh-7 cells were co-transfected with a plasmid carrying the HDV WT genome or the different mutants described, along with a plasmid carrying the HBV 1.3× genome [30]. After transfection, cells were harvested on days 1, 3, 7, 10, and 14 for the analysis of HDAg expression by Western blot and on days 7 and 14 for RNA extraction to determine and quantify the presence of HDV genomes by RT-qPCR.

As shown in Figure 1B, in cells transfected with WT HDV on day 1, we detected S-HDAg, and by day 3, and more clearly on day 7, we detected both S-HDAg and L-HDAg, as previously described [30,31]. No protein expression was detected in cells transfected with the HDV mutants in which Ser177 was mutated (HDV-Ser177Ala or HDV-Ser177Asp) or in the one where we introduced a mutation that altered ribozyme conformation (Figure 1B). In cells transfected with HDV-NPrL-HDAg, S-HDAg was clearly detected from day 1 and remained stable with time (the decrease observed on day 10 was due to a protein loading problem). However, L-HDAg expression levels was very low in comparison to HDV WT (Figure 2B). Cells transfected with HDV-ΔNLS-HDAg or HDV-ΔNES-HDAg mutants showed a very similar kinetic of HDAg expression to the one shown by WT, but the levels of L-HDAg expression were lower than in WT.

The analysis of the presence of HDV genomes showed no HDV replication activity in cells transfected with HDV mutants in which Ser177 was mutated, nor in the ribozyme mutant. The levels of HDV replication were significantly lower in the cells transfected with the HDV-NPrL-HDAg in comparison to WT at 7 dpt. In contrast, they were significantly higher than in WT with the HDV-ΔNLS-HDAg or HDV-ΔNES-HDAg mutants in comparison to WT on day 7, which might be explained by the lower L-HDAg. However, on day 14, the differences among the groups disappeared (Figure 1C).

### 3.2. Analysis of Intracellular Localization of HDAgs and Assessment of HDV Infectious Virion Formation

On days 7 and 14 post-transfection of HDV-NPrL-HDAg, HDV-ΔNLS-HDAg, and HDV-ΔNES-HDAg mutants, Huh-7-cells were fixed, and HDAg expression was analyzed by immunofluorescence. As shown in Figure 2A, transfection with the plasmid carrying WT HDV resulted in a mainly nuclear presence of HDAg on day 7, and by day 14, expression could be detected both in the nucleus and the cytoplasm. In cells transfected with NPrL-HDAg, both on day 7 and day 14, the antigen was predominantly detected in the nucleus, reflecting the lower levels of L-HDAg, which is required for nuclear export. The mutant expressing ΔNLS-HDAg exhibited a distinct cytosolic presence on day 7, contrasting with the predominantly nuclear location observed in HDV WT. By day 14, the expression significantly diminished in both the nucleus and cytoplasm. In contrast, cells transfected with HDV-ΔNES-HDAg displayed a robust and nearly exclusive nuclear localization on both day 7 and day 14 (Figure 2A).

Interestingly, we observed that the morphology of the nucleus of HDV-NPrL-HDAg- transfected cells was clearly altered. In fact, the analysis of the size of the transfected cells with the different HDV variants revealed a significant nuclear enlargement in the cells transfected with NPrL-HDAg with respect to cells transfected with HDV WT or the other two mutants (Figure 2B).

Then, we analyzed the formation of HDV infectious particles in the supernatant of transfected cells. For this purpose, cell supernatant was harvested on days 7 and 14 and used to infect Huh-7 cells stably expressing hNTCP. A minimal number of infected cells were observed when cells were incubated with day 7 supernatants, but the count significantly increased when using day 14 HDV-WT supernatants. While infectious particles were detected in the supernatant from ΔNLS-HDAg transfected cells, none were observed in the supernatants of NPrL-HDAg- or ΔNES-HDAg-transfected cells (Figure 2C).

### 3.3. In Vivo Analysis of HDV-NPrL-HDAg, HDV-ΔNLS-HDAg, and HDV-ΔNES-HDAg Mutants

For in vivo evaluation, liver-tropic AAV serotype 8 vectors carrying HDV WT or the indicated mutants were generated. Subsequently, the AAV-HDV vectors were intravenously administered to C57BL/6 male mice along with AAV-HBV at a dose of 5 × 10^10^ vg/mouse each, and the animals were sacrificed three weeks later (Figure 3A). Upon sacrifice, the liver was extracted, and the presence of HDAg was analyzed through Western blot and immunohistochemistry, while HDV genomes were assessed via RT-qPCR.

The Western blot analysis revealed the expression of both S-HDAg and L-HDAg in animals receiving HDV-WT and the different mutants (Figure 3B). However, significant differences were observed in the L-HDAg/S-HDAg ratio depending on the mutant. The ratio was lower in HDV-NPrL-HDAg and HDV-ΔNES-HDAg and higher in HDV-ΔNLS-HDAg compared to HDV WT (Figure 3C). Immunohistochemistry analysis of HDAg in the liver of HDV WT revealed the expected nuclear-cytoplasmic distribution, with weak cytoplasmic staining. In the animals receiving the HDV-NPrL-HDAg and HDV-ΔNES-HDAg mutants in addition to nuclear staining, we observed a strong cytoplasmic staining in some hepatocytes. However, a considerable number of cells exhibited almost exclusive nuclear staining, which was weaker in HDV-ΔNES-HDAg-injected mice (Figure 3D). Moreover, animals receiving HDV-ΔNLS-HDAg displayed a very intense cytoplasmic signal, correlating with higher L-HDAg/S-HDAg ratios (Figure 3D).

HDV genomes were significantly higher in the mice receiving HDV WT than in the animals receiving the different mutants (Figure 3E). This suggests deficiencies in HDV replication associated with the alteration of HDAg intracellular location or a mutant-related toxic effect.

### 3.4. Evaluation of Liver Damage

To evaluate liver damage, we separated the study into two different experimental groups. The first group was composed of HBV HDV WT-, HBV- NPrl Ag-HDV-injected, and untreated control animals, and the second group was composed of HBV-HDV WT-, HDV-ΔNLS-HDAg-, and HDV-ΔNES-HDAg-treated mice. Transaminase levels were analyzed on days 7, 14, and 21 after vector injection, as described in Figure 3A. As anticipated in both experiments, AAV-HDV WT mice exhibited ALT elevation above normal levels in both experiments. Transaminase levels were also elevated in animals receiving the different mutants (Figure 4A). However, while in mice receiving HDV-ΔNLS-HDAg or HDV-ΔNES-HDAg the levels were similar to that in HDV WT, in animals receiving HDV-NPrL-HDAg, the ALT levels were significantly higher on day 21. The pattern of transaminase elevation was very similar in all cases, showing an increase from day 7 to day 21, except for HDV-ΔNLS-HDAg, where the peak was achieved on day 14 (Figure 4A).

Due to the elevated transaminase levels in HDV-NPrL-HDAg, we conducted a more detailed analysis of the liver pathology in those mice. H&E staining revealed a more altered liver architecture in HDV-NPrL-HDAg than in HDV WT mice (Figure 4B). The analysis of the size of hepatocyte nuclei revealed, as previously observed, bigger nuclei in HDV WT animals compared to controls, which were even larger in HDV-NPrL-HDAg mice (Figure 4C). Previously, we demonstrated in this animal model that HDV-induced liver damage was associated with the induction of apoptosis, identified by the presence of activated Caspase 3 (a-Casp3), and an increase in parenchymal macrophages and TNF-α expression [30]. Here, we observed that HDV-NPrL-HDAg increased the number of apoptotic hepatocytes compared to HDV-WT (Figure 4D), and we also observed a stronger inflammatory macrophage infiltrate (Figure 4E,F) and significantly higher TNF-α production (Figure 4G). The analysis of the expression of additional cytokines revealed a significant increase in TGF-β and a tendency to increase in IFN-γ and IL-6 (Figure 4G). Interestingly, IFN-β, which is strongly induced by HDV, was significantly lower in HDV-NPrL-HDAg (Figure 4G) probably as a consequence of lower genome levels (Figure 3E).

## 4. Discussion

In this study, we comprehensively assessed in vitro and in vivo the consequences of introducing various mutations into the HDV genome, specifically targeting HDAg intracellular localization, post-translational modification, and the functionality of the HDV ribozyme. Utilizing a surrogate system developed in our laboratory, involving plasmids and a liver-tropic AAV vector carrying the HDV 1.2× sequence, developed by Dr. Taylor’s group [38], under the transcriptional control of a liver-specific promoter, we achieved transcription of the HDV anti-genome sequence. This sequence initiated HDV replication and the expression of the two HDAgs, in cell culture and, more importantly, in the livers of mice [30].

Our in vitro investigations were conducted in the human hepatic cell line Huh-7, known for sustaining HDV replication due to the absence of a type I response [31]. We observed that mutations in the Ser177 residue, caused either by replacing it with the neutral amino acid alanine or the phosphomimetic amino acid aspartic acid, resulted in the complete absence of protein expression and a substantial impact on HDV replication. Ser177 phosphorylation of S-HDAg is crucial for viral replication, as it is required for interaction with cellular RNA Pol II in the production of new genomes [19]. Thus, unphosphorylated S-HDAg cannot interact with the host polymerase, preventing the replication of the HDV antigenome. Furthermore, the results obtained from transfection with the mutant, in which the Ser residue was replaced with the phosphomimetic amino acid Asp, underscore the importance of maintaining a delicate equilibrium between unphosphorylated and phosphorylated S-HDAg at Ser177 in the HDV life cycle. Indeed, it has been demonstrated that only unphosphorylated S-HDAg is incorporated into HDV particles [22]. Our data confirm the importance of maintaining a proper balance between phosphorylated and unphosphorylated S-HDAg at Ser177 for the correct establishment of the HDV life cycle.

Similar outcomes were observed following the transfection of the HDV mutant, where we introduced a conformational change in the ribozyme sequence of the antigenome. Drawing on previous studies involving the truncation of different regions of the HDV ribozyme, we engineered an HDV mutant with a mutation in helix 1 of the antigenomic HDV ribozyme. The transfection of this mutant into Huh-7 cells resulted in no HDAg production or HDV replication, confirming the anticipated importance of the correct folding of this sequence for establishing HDV replication [9]. Due to the lack of protein expression and HDV replication in vitro, Ser177 and ribozyme mutants were not evaluated in vivo.

Regarding the non-prenylated mutant, we observed an altered L-HDAg/S-HDAg ratio both in vitro and in vivo through Western blot (WB) analysis. While normal levels of S-HDAg expression were detected, the expression of non-prenylated L-HDAg was significantly lower. These data suggest that the lack of isoprenylation might affect the stability of L-HDAg, resulting in a lower L-HDAg/S-HDAg ratio compared to HDV WT. Alternatively, the absence of prenylated L-HDAg might impact HDV genome editing, reducing the expression of this antigen. However, recent findings by Verrier et al. demonstrated that treating HepaRG cells with an isoprenylation inhibitor resulted in the accumulation of edited HDV genomes, reducing the likelihood of this hypothesis [39]. Furthermore, our in vitro experiments clearly indicated that isoprenylation plays a crucial role in the cellular localization of HDAgs, leading to a predominant nuclear localization. This observation was also noted in some mouse hepatocytes. The absence of isoprenylation likely disrupts the normal translocation of HDAgs to the cytoplasm, resulting in its nuclear retention. Similar observations were made previously in HDV mutants lacking L-HDAg [31] and here in the HDV-ΔNES-HDAg, highlighting the role of L-HDAg in the translocation of HDV RNPs from the nucleus to the cytoplasm.

We also found that the lack of isoprenylation leads to a delay in viral replication in vitro and a significant reduction in viral replication in vivo. This suggests that this translational modification might play a role in the regulation of HDV replication. Interestingly, according to a previous publication, an increase in viral replication would be expected, since earlier findings attributed the inhibition of viral replication to prenylated L-HDAg; however, we observed the opposite situation [40]. One hypothesis explaining the decreased replicative capacity could be attributed to impaired HDAg localization and altered membrane association [41]. Isoprenylation facilitates the association of HDV ribonucleoproteins (RNPs) with cellular membranes. Without isoprenylation, this association may be compromised, affecting the interaction of HDAgs with cellular kinases [19,20,21,22]. As expected, this mutant was not capable of producing HDV infectious particles, as this motif is essential for the interaction with HBsAg. In summary, inhibiting the isoprenylation of L-HDAg has multifaceted effects on the composition, stability, cellular localization, and membrane association of HDAgs, ultimately impacting the efficiency of HDV replication.

However, our more important finding associated with this mutant is its association with exacerbated liver pathology. We observed, both in vitro and in vivo, a higher enlargement of the nuclei compared to HDV-WT. Nuclear enlargement is indicative of cellular stress and toxicity that might be associated with alterations in cellular processes, including damage to the DNA, disruption of normal cellular functions, or induction of cellular stress responses [42]. Even though the replicative capacity of this mutant was lower than the WT, it was able to induce stronger liver damage than the HDV WT vector, as evidenced by significantly higher transaminase elevation and a higher number of apoptotic hepatocytes. This higher toxicity was accompanied by a stronger macrophage infiltrate and higher expression of proinflammatory cytokines such as TNF-α or TGF-β. These results are in line with a previous study that demonstrated a link between TNF-α and HDV-induced liver toxicity, as the administration of the TNF-α inhibitor etanercept resulted in the amelioration of liver injury [32]. Interestingly, the expression of type-I IFN was lower than the one in the HDV WT, most likely indicative of the lower replication capacity of the mutant; however, it was in contradiction to the observation of cell culture, where treatment with isoprenylation inhibitors resulted in an increase in IFN-β production [43].

The higher toxicity of the non-prenylated mutant could be associated with the altered L-HDAg/S-HDAg [31]; we showed previously that an HDV mutant lacking L-HDAg expression, only expressing S-HDAg, was more toxic than the HDV WT, and that this toxicity can be reduced by L-HDAg complementation. These data point toward S-HDAg as the antigen preferentially involved in HDV cytotoxicity [31,44]. However, it can not be rule out that the unprenylated L-HDAg interfered with host processes. As we know, the elimination of the isoprenylation site is detrimental to viral assembly; consequently, the HDV RNPs accumulate inside the cells [45]. The intracellular/intranuclear accumulation of the HDAgs could impair biological functions and induce cell stress or hepatocyte death, as has been described for other viral proteins. One example is HCV, which induces ER stress and mitochondrial alterations through calcium signaling, leading to the production of ROS that are normally found in chronic HCV patients [46,47]. In the case of HBV, the HBx protein promotes the activation of the inflammasome, the production of mitochondrial ROS, and the induction of pyroptosis in the infected cell [48]. Additional studies are needed to clarify the role of the isoprenylated and non-isoprenylated L-HDAg within the HDV-induced pathology.

Next, we determined the role of the intracellular location of HDAgs in the HDV viral cycle. For that purpose, we altered the nuclear localization signal of HDAgs by the substitution of residues Glu-66 and Arg-75 by Ala, reported as essential for the correct functionality of the NLS domain [13,14,15]. The HDAgs expressed by the HDAg-ΔNLS mutant displayed a subcellular distribution pattern different from that of the HDV WT, with a predominant cytoplasmic localization of the HDAgs. However, this mutation does not eliminate the presence of the HDAgs in the nucleus, as shown both in Huh-7 cells and mice, indicating that other sequences should be involved in the traffic of the antigens from the cytoplasm to the nucleus. The discrepancy between our results and those of Alves et al. could be due to differences in the HDV constructs, since they deleted residues 66 and 75 while we substituted them with alanines [15]. Therefore, the identification of more residues implicated in the functionality of the NLS domain located between aa 66 and 75 should be considered. Furthermore, this mutant showed a decrease in HDV replication that can not be attributed to the absence of HDAg in the nucleus. Alternatively, it could be attributed toan additional function of residue Arg-75, as it has been described to be essential for the proper recruitment of host RNA Pol by binding to chromatin-remodeling complexes [49,50].

Finally, we have generated a mutant in which we altered the capacity of the L-HDAg to exit the nucleus by mutating the NES signal. The NES domain is located exclusively in the L-HDAg between the residues 198–210 (ILFPADPPFSPQS). In particular, Pro-205 was demonstrated to be essential for the translocation of HDV RNP from the nuclear to the cytosol [16]. Based on those studies, we constructed an HDV mutant with a truncated NES by replacing Pro-205 with Ala-205. Despite the loss of nuclear export, HDAg-staining was detected in the cytoplasmic compartment in both transfected Huh-7 cells and C57BL/6 mice, so some amount of protein is retained after synthesis.

Although the replication capacity of this mutant in vitro was similar to or higher than that shown by HDV-WT in vitro, it was unable to form HDV infectious particles. These results indicate that once L-HDAg is inside the nucleus, if the NES domain is not functional, the RNP particles can not be exported to the cytoplasm, and HDV infectious particles are not formed. Furthermore, the trafficking of HDAgs from the nucleus to the cytoplasm is necessary for the interaction between L-HDAg and HBsAg in the ER and, consequently, is required for the production of infectious particles. These findings confirm previous data reported by Lee et al., demonstrating that this residue within the NES domain is indispensable for the nuclear export of HDV RNPs mediated by L-HDAg [16]. In mice, we observed that the lack of NES altered the ratio of L-HDAg/S-HDAg and the replication capacity of HDV, which in both cases was lower than in HDV WT. The liver toxicity associated with the administration of this mutant was similar to the one of HDV-WT.

## 5. Conclusions

Taken together, our results highlight the relevance of HDAgs and their posttranslational modification in HDV biology and HDV-induced liver injury, demonstrated for the first time in an animal model. The increased toxicity of the mutant expressing a non-prenylated L-HDAg correlates with a higher macrophage infiltrate and TNF-α expression. Once again, our data showcase the versatility of the AAV-based HDV model in addressing fundamental questions in HDV biology in vivo.

## Figures and Tables

**Figure 1 viruses-16-00379-f001:**
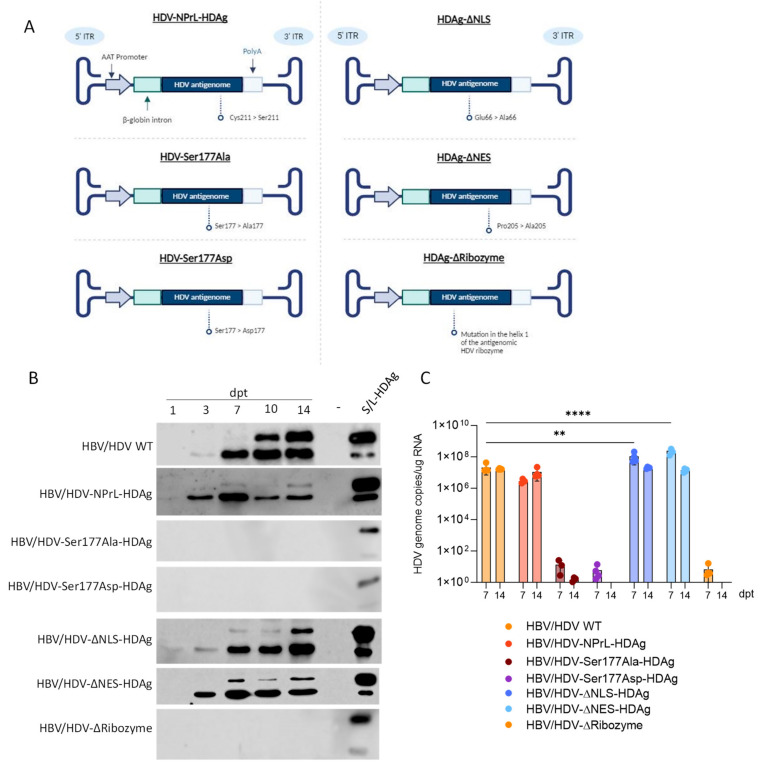
The analysis of HDAg expression and HDV genomes in Huh-7 cells transfected with HDV-WT or the different mutants. (**A**) A schematic representation of the different mutants used in the study. The transcription of the HDV antigenome is regulated by the human α-1-antitrypsin promoter (AAT), and flanked by a β-globin intron and a poly-adenylation signal sequence. Specific mutations are indicated. (**B**) Huh-7 cells were transfected with HDV WT plasmid or the different mutants indicated in the figure together with a plasmid carrying HBV 1.3× genome. Cells were harvested on days 1, 3, 7, 10, and 14 post-transfection (dpt), and HDAg expression was analyzed by Western Blot in cell lysates. As control Huh-7 cells were co-transfected with plasmids expressing S-HDAg or L-HDAg under the control of a liver-specific promoter and collected on 3 dpt [30]. (−) indicates non-transfected cells. (**C**) The cells were transfected as described in B and harvested on 7 and 14 dpt. Total RNA was extracted, and HDV genome levels were quantified by RT-qPCR and normalized using total RNA concentration. Statistical analysis was performed using a two-way ANOVA followed by Dunnett’s multiple comparison test (** *p* < 0.01, **** *p* < 0.0001).

**Figure 2 viruses-16-00379-f002:**
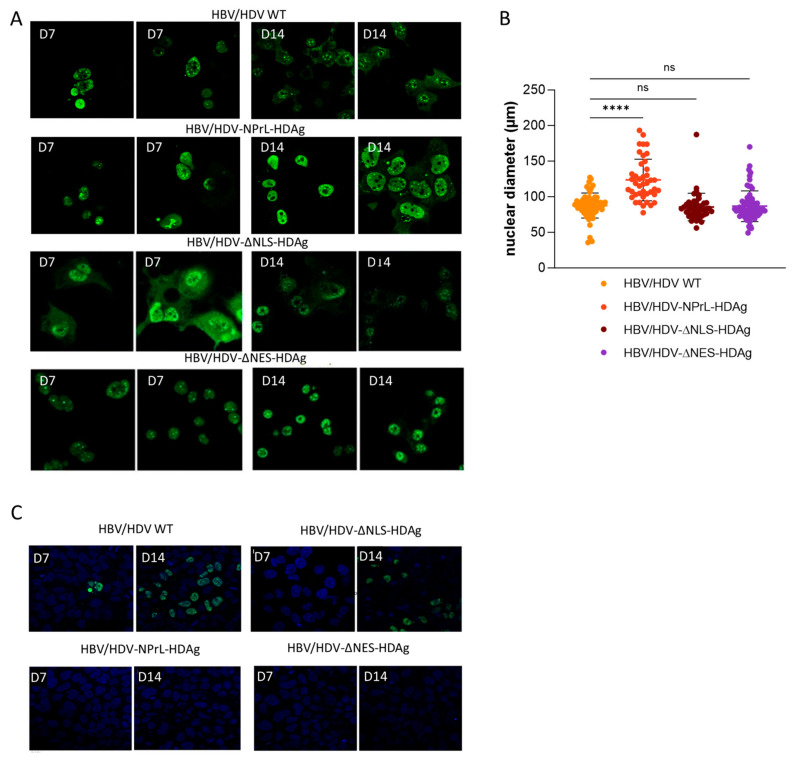
An analysis of HDAg intracellular localization in Huh-7 cells transfected with HDV-WT or the different mutants and an evaluation of the capacity to produce HDV infectious particles. (**A**) HDAg localization was examined by immunofluorescence (IF). (**B**) The nuclear diameter of HDAg-positive hepatocytes was measured from IF images using Aperio Image Scope v12.3 software with the ruler tool, at 20× magnification. Statistical analysis was performed using ordinary one-way ANOVA (**** *p* < 0.0001). (**C**) Huh-7.5.1-hNTCP cells were infected with supernatants from co-transfected cells collected on 7- and 14-dpt. On day 7 post-infection, the cells were fixed and immune-stained with anti-HDV human serum for HDAg detection (ns, non-significant).

**Figure 3 viruses-16-00379-f003:**
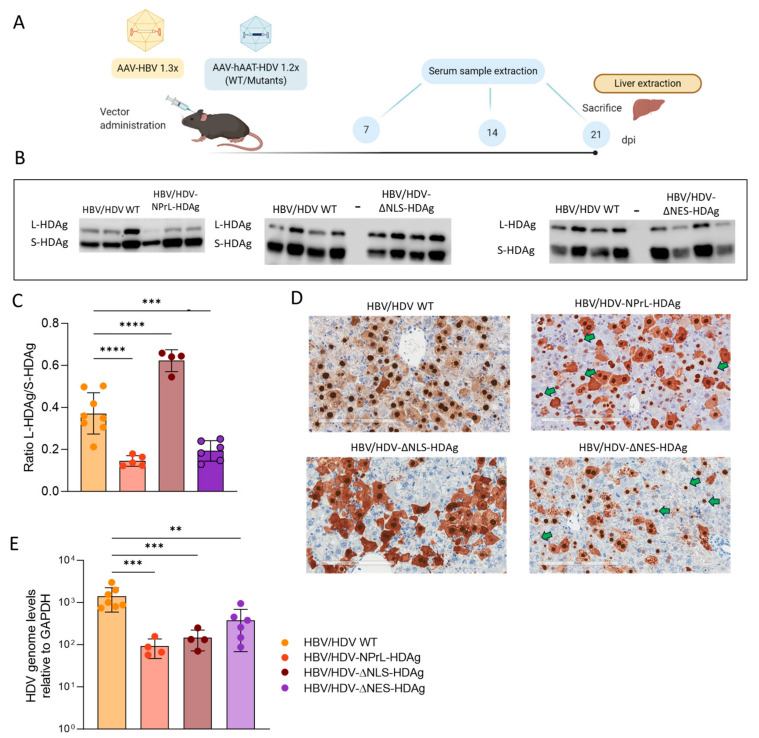
An in vivo evaluation of HDV mutants. (**A**) A schematic representation of the experimental layout. C57BL/6 male mice intravenously received AAV HDV or AAV HDV mutants along with AAV-HBV, at a dose of 5 × 10^10^ vg/mouse each. Serum samples were collected on days 7, 14, and 21 post-injection (dpi), and the animals were sacrificed in week 3, with the liver harvested. (**B**) A Western blot analysis of liver lysates was performed to detect S-HDAg and L-HDAg. (**C**) HDAg expression levels were quantified, and the L-HDAg/S-HDAg ratio was calculated for each animal. (**D**) Immunostaining against HDAgs was conducted on 21 dpi in the liver sections of mice injected with AAV-HDV WT or the different mutants. Green arrows indicate hepatocytes where exclusive nuclear staining was detected while cytoplasmic staining was undetectable or very weak. Scale bar: 200 μm. (**E**) Total RNA was extracted from the livers, and HDV genome levels were assessed by RT-qPCR and normalized using GAPDH as a housekeeping gene. Statistical analysis was performed using ordinary one-way ANOVA followed by multiple comparisons (** *p* < 0.01, *** *p* < 0.001, **** *p* < 0.0001).

**Figure 4 viruses-16-00379-f004:**
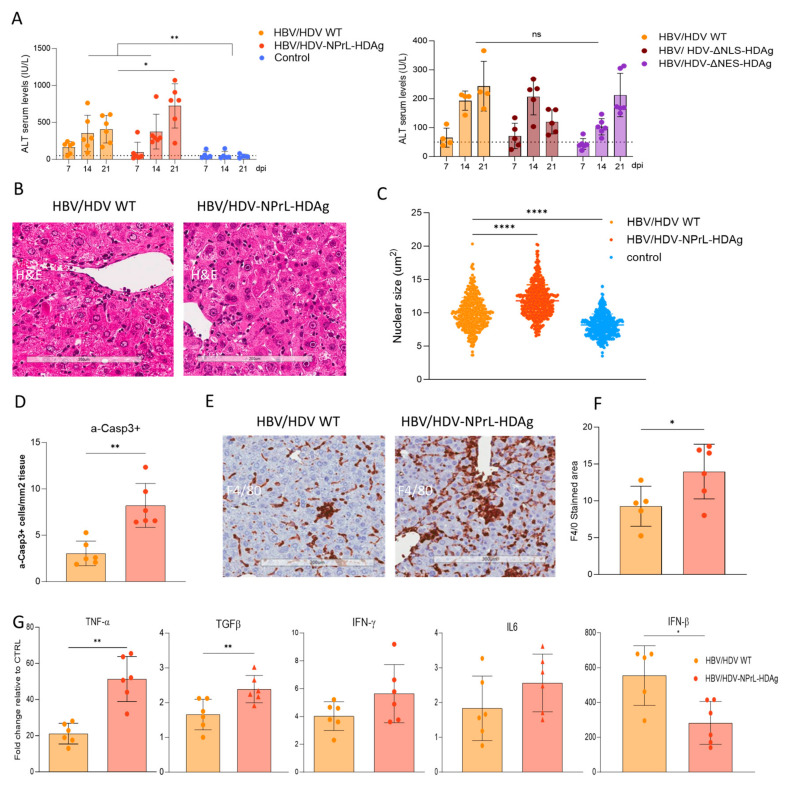
An analysis of liver pathology. (**A**) Peripheral blood was collected every 7 days for 3 weeks to measure ALT concentration in the serum (as described in Figure 3A). Individual data points and mean values ± standard deviation were plotted, and significant differences between groups at each time point were determined by a mixed-effect analysis and multiple comparisons (* *p* < 0.05, ** *p* < 0.01). (**B**) Liver sections from HBV/HDV WT- and HBV/HDV-NPrL-HDAg-injected mice obtained on 21 dpi were analyzed by H&E staining, revealing more abundant degenerated nuclei in the HDV-NPrL-HDAg mutant. Scale bar: 200 μm. (**C**) The nuclear diameter was significantly larger in the HDV-NPrL-HDAg group, as determined by an unpaired *t*-test (**** *p* < 0.0001). (**D**) Immunostaining against activated caspase 3 (a-casp3) revealed that the HDV-NPrL-HDAg mutant exacerbated hepatocyte death by apoptosis, with differences between the two groups revealed by a Mann–Whitney test. (**D**–**F**) Liver sections were stained with anti-F4/80 to determine the presence of macrophages, showing a significantly higher presence in the HDV-NPrL-HDAg group. Scale bar: 200 μm (* *p* < 0.05, ** *p* < 0.01). (**G**) The expression of different cytokines in the liver of HBV/HDV WT and HBV/HDV-NPrL-HDAg was analyzed by RTqPCR and compared to those in untreated control animals, revealing differences between the two groups (* *p* < 0.05, ** *p* < 0.01). A Mann–Whitney test was applied for the statistical comparison of the two groups.

**Table 1 viruses-16-00379-t001:** A description of the primers used for cloning the indicated HDV mutants.

HDV Mutant	Primer	Sequence	Kit
HDV-NPrL-HDAg	Fw	5′-TTTTCTCCCCAGAGTTCTCGACCCCAGTGAATAAAG-3′	SDM (QuikChange II)
	Rv	5′-CTTTATTCACTGGGGTCGAGAACTCTGGGGAGAAAA-3′	
HDV-Ser177Ala	Fw 1	5′-ACTCCGGACCTGGGAAGAGGCCTCTCAGGGGAGGATTCAC-3′	SDM (InFusion Cloning)
	Rv 1	5′-TCCGAGAGAAGGGGGCCTCCGGGA-3′	
	Fw 2	5′-AGGGAGTCCCGGAGGCCCCCTTCTCTC-3′	
	Rv 2	5′-GAGCAGCGCTGCTCGAGGCAAGCTTGCATGCCTGCAGGTC-3′	
HDV-Ser177Asp	Fw 1	5′-ACTCCGGACCTGGGAAGAGGCCTCTCAGGGGAGGATTCAC-3′	SDM (InFusion Cloning)
	Rv 1	5′-GGTCCGAGAGAAGGGGTCCTCCGGGA-3′	
	Fw 2	5′-AGGGAGTCCCGGAGGACCCCTTCTCTC-3′	
	Rv 2	5′- GAGCAGCGCTGCTCGAGGCAAGCTTGCATGCCTGCAGGTC-3′	
HDV-ΔRibozyme	Fw 1	5′-CAAAGAATTGGGATTCGAACATCGATTGAATTCCCCGGGGA-3′	SDM (InFusion Cloning)
	Rv 1	5′-TCTTACCTGATGGGGCTCATGGTCCCA-3′	
	Fw 2	5′-GAGGCTGGGACCATGAGCCCCATCAGGTAA-3′	
	Rv 2	5′-AGCAAAGAAAGCAACGGGGCTAGCCGGTGGGTGTTC-3′	
HDAg-ΔNLS	Fw 1	5′-GGGGGCGGAACACCCACCGGCTAGCCCCGTTGCTTTCTTT-3′	SDM (InFusion Cloning)
	Rv 1	5′-TCGCCGGGGGAGCCCCTGCTCCATCCTTATCCTT-3′	
	Fw 2	5′-AAGGATAAGGATGGAGCAGGGGCTCCCC-3′	
	Rv 2	5′-GAGCAGCGCTGCTCGAGGCAAGCTTGCATGCCTGCAGGTC-3′	
HDAg-ΔNES	Fw 1	5′-ACTCCGGACCTGGGAAGAGGCCTCTCAGGGGAGGATTCAC-3′	SDM (InFusion Cloning)
	Rv 1	5′-TGGGGAGAAAAGGGTGCATCGGCT-3′	
	Fw 2	5′-TCTTCCCAGCCGATGCACCCTTTTCT-3′	
	Rv 2	5′-CTCTCGAGCAGCGCTGCTCGAGGCAAGCTT-3′	

## Data Availability

The original contributions presented in the study are included in the article material; further inquiries can be directed to the corresponding authors.

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
