# Peer review of "Deciphering the Role of Post-Translational Modifications and Cellular Location of Hepatitis Delta Virus (HDV) Antigens in HDV-Mediated Liver Damage in Mice"

_viruses, 2024, doi:10.3390/v16030379_

Round 1

Reviewer 1 Report

Comments and Suggestions for Authors

This paper studied post translationnal modifications and and location of HDAgs in mice  

Most findings confirm previous authors reports and literature datas

One  major issue is that mice toxicity is quite different from histopthological changes in man casting  doubts on the relevance of the findings 

authors should  adress this in details to improve the impact and relevance  of the work 

Comments on the Quality of English Language

moderate editing by native english expert 

Author Response

Point by point answer to reviewer 1

This paper studied post translationnal modifications and and location of HDAgs in mice. Most findings confirm previous authors reports and literature data

We agree with the reviewer most of our data describe aspects of HDV virology previously published, however, all the experiments were performed in vitro and in our study we were able to test the different mutants in the liver of mice.

One  major issue is that mice toxicity is quite different from histopthological changes in man casting  doubts on the relevance of the findings

We respectfully disagree with the reviewer's assessment. Mice receiving AAV-HBV/HDV developed hepatic lesions characterized by diffuse hepatocyte hypertrophy, inflammation with piecemeal necrosis, and focal/multifocal single-cell necrosis. These features have been observed in biopsy specimens of patients with chronic HDV infection (1-3). References included in the text (33-35).

  1. Buitrago B, Popper H, Hadler SC, Thung SN, Gerber MA, Purcell RH, et al. Specific histologic features of Santa Marta hepatitis: a severe form of hepatitis d-virus infection in Northern South America. Hepatology 1986;6(6):1285–1291.
  2. Verme G, Amoroso P, Lettieri G, Pierri P, David E, Sessa F, et al. A histological study of hepatitis delta virus liver disease. Hepatology 1986;6(6):1303–1307
  3. Sagnelli, E.; Felaco, F.M.; Filippini, P.; Pasquale, G.; Peinetti, P.; Buonagurio, E.; Aprea, L.; Pulella, C.; Piccinino, F.; Giusti, G. Influence of HDV infection on clinical, biochemical and histological presentation of HBsAg positive chronic hepatitis. Liver. 1989, 9, 229–234.

English has been reviewed

Reviewer 2 Report

Comments and Suggestions for Authors

General comment

The paper does not provide much new insight into the molecular biology of HDV, but the transduction of HDV mutants to susceptible mice via an AAV-vector is indeed novel and resulted in new knowledge on the pathology of hepatitis delta. The infection system may be useful for development of therapies against HDV. The text is overall very well written, somewhat lengthy in parts. The figures are instructive and support the data and conclusions.

Some minor points should be corrected in a revision.

1.      Title. Spell out HDV in title

2.      Typos. L29: od, grammar and later there are also some missing words or typos.

3.      L99. For which words stands HV?

4.      L141. 293T cells are fibroblasts and not hepatoma cells.

5.      L154. Name the supplier of Optimem.

6.      L221. Explain here the abbr. PFA, not later in L243.

7.      Fig. 1A is unsatisfactory. Neither is the expression vector explained, nor is a map of the two HDV proteins with their domains and the exact sites of the mutations shown.

8.      Fig. 3D. The text in L372-376 is not agreeable. “In the animals receiving the HDV-NPrL-HDAg and 372 HDV-ΔNES-HDAg mutants, a very similar pattern to HDV WT was observed, with the majority of hepatocytes showing cytoplasmic-nuclear staining, the cytoplasmic staining was particular intense in HDV-NPrL-HDAg. However, a considerable number of cells exhibited almost only nuclear staining (Figure 3D)”

a.      The cells infected with HDV-NPrL-HDAg or HDV-ΔNLS-HDAg show in addition to nuclear staining a much stronger cytoplasmic staining than the HDV WT.

b.      The nuclear staining of HDV ΔNES showed a much weaker nuclear staining than WT.

9.      L428. There is no panel H of fig. 4.

10.   L490-492. “Furthermore, our in vitro experiments clearly indicated that isoprenylation plays a crucial role in the cellular localization of HDAgs, leading to a predominant nuclear localization.” The other parts of the text (L492-494) suggest that the authors mean the opposite, i.e., cytoplasmic.

Comments on the Quality of English Language

Overall is the English very good. Some grammatical errors with plural and singular occur. Some words are missing

Author Response

We are very grateful to the referee for the comments to improve the manuscript and support for publication.

  1. Title. Spell out HDV in title

HDV has been spell out in the title

  1. Typos. L29: od, grammar and later there are also some missing words or typos.

Corrected

  1. L99. For which words stands HV?

It’s is a mistake it should be HDV. corrected

  1. L141. 293T cells are fibroblasts and not hepatoma cells.

Sorry for this, we have introduced a better description of the cell line.

  1. L154. Name the supplier of Optimem.

The information has been included.

  1. L221. Explain here the abbr. PFA, not later in L243.

Corrected.

  1. Fig. 1A is unsatisfactory. Neither is the expression vector explained, nor is a map of the two HDV proteins with their domains and the exact sites of the mutations shown.

We have changed Figure 1A including all the elements present in the construct as the localization of the mutants.

  • Fig. 3D. The text in L372-376 is not agreeable. “In the animals receiving the HDV-NPrL-HDAg and 372 HDV-ΔNES-HDAg mutants, a very similar pattern to HDV WT was observed, with the majority of hepatocytes showing cytoplasmic-nuclear staining, the cytoplasmic staining was particular intense in HDV-NPrL-HDAg. However, a considerable number of cells exhibited almost only nuclear staining (Figure 3D)
  • The cells infected with HDV-NPrL-HDAg or HDV-ΔNLS-HDAg show in addition to nuclear staining a much stronger cytoplasmic staining than the HDV WT.
  • The nuclear staining of HDV ΔNES showed a much weaker nuclear staining than WT.

We have modified the text for a more accurate description of the results.

  1. L428. There is no panel H of fig. 4.

Corrected

  1. L490-492. “Furthermore, our in vitro experiments clearly indicated that isoprenylation plays a crucial role in the cellular localization of HDAgs, leading to a predominant nuclear localization.” The other parts of the text (L492-494) suggest that the authors mean the opposite, i.e., cytoplasmic.

We have modified the text for a more accurate description of the results.

Round 2

Reviewer 1 Report

Comments and Suggestions for Authors

AUTHORS did replied to most queries  and the paper is now suitable for publication